# Phage peptides mediate precision base editing with focused targeting window

Kun Jia[1,2,3,8], Yan-ru Cui[1,4,8], Shisheng Huang[1,4], Peihong Yu[1,4], Zhengxing Lian[5], Peixiang Ma[1] & Jia Liu [1,2,3,6,7✉]

Base editors (BEs) are genome engineering tools that can generate nucleotide substitutions without introducing double-stranded breaks (DSBs). A variety of strategies have been developed to improve the targeting scope and window of BEs. In a previous study, we found that a bacteriophage-derived peptide, referred to as G8P$_{PD}$, could improve the specificity of Cas9 nuclease. Herein, we investigate the applicability of G8P$_{PD}$ as molecular modulators of BEs. We show that G8P$_{PD}$ can improve cytidine base editor (CBEs) and adenine base editor (ABE) to more focused targeting windows. Notably, in a cell-based disease model, G8P$_{PD}$ increases the percentage of perfectly edited gene alleles by BEs from less than 4% to more than 38% of the whole population. In addition, G8P$_{PD}$ can improve the targeting scope of BE in mouse embryos. In summary, our study presents the peptidyl modulators that can improve BEs for precision base editing.

[1] Shanghai Institute for Advanced Immunochemical Studies and School of Life Science and Technology, ShanghaiTech University, Shanghai 201210, China. [2] The State Kay Laboratory of Respiratory Disease, First Affiliated Hospital of Guangzhou Medical University, Guangzhou 510120, China. [3] Shanghai Clinical Research and Trial Center, Shanghai 201210, China. [4] University of Chinese Academy of Sciences, Beijing 100049, China. [5] Beijing Key Laboratory of Animal Genetic Improvement, China Agricultural University, 2 Yuanmingyuan West Rd., Haidian District, Beijing 100094, China. [6] Gene Editing Center, School of Life Science and Technology, ShanghaiTech University, Shanghai 201210, China. [7] Guangzhou Laboratory, No. 9 XingDaoHuanBei Road, Guangzhou International Bio Island, Guangzhou 510005 Guangdong Province, China. [8] These authors contributed equally: Kun Jia, Yan-ru Cui. ✉email: liujia@shanghaitech.edu.cn

Clustered regularly interspaced short palindromic repeats (CRISPR)-CRISPR-associated genes (Cas) is the bacterial adaptive immune system for protecting host organisms from invading pathogens[1–3]. Owing to the modular feature, type II CRISPR systems, particularly CRISPR-Cas9, have been widely used for genome editing, transcriptional and epigenetic modulation and molecular imaging[4]. CRISPR-Cas9 can be directed to human genome by single-guide RNA (sgRNA) to create double-stranded breaks (DSBs) at targeted genomic loci[5–8]. In human cells, DSBs are repaired by two competing DNA repair pathways: error-prone nonhomologous end joining (NHEJ) and homology directed repair (HDR). The latter repair pathway can facilitate CRISPR-Cas9-mediated gene correction of pathogenic mutations[4].

Recent studies have highlighted base editors (BEs) as an efficient genome editing tool for precision gene therapy[9–11]. BEs are fusion proteins comprising a catalytically inactive Cas nuclease and a nucleobase deaminase[12,13]. Unlike CRISPR-Cas9 genome editing tools, BEs generate precise base substitutions without introducing DSBs, thus avoiding concurrent, competing NHEJ events that incorporate nucleotide insertions and deletions (indels)[14]. Cytidine base editors (CBEs) and adenine base editors (ABEs) have been developed to realize genomic alterations of C-G to T-A and A-T to G-C, respectively. Besides DNA editing, Cas variants with RNA-binding capacities have been adapted for programmable RNA editing[15,16].

The widely used third-generation CBEs, referred to as BE3, are fusion proteins composed of rat APOBEC (rAPOBEC), Cas9 D10A nickase and uracil glycosylase inhibitor (UGI). BE3 has a five nucleotide targeting window ranging from position 4 to 8, counting the protospacer adjacent motif (PAM) trinucleotides as positions 21 to 23[12]. An engineered CBE variant A3A, where rAPOBEC is replaced with human APOBEC3A (hAPOBEC3A), has a wider targeting window at positions 2 to 13 and is capable of editing methylated genomic regions[17]. Engineering endeavors have been made to improve the editing window of both BE3 (rAPOBEC-nCas9-UGI)[18] and A3A (hAPOBEC3A-nCas9-UGI) CBEs[19]. Compared with ABEs, CBEs have higher genome-wide off-target events[20,21]. Despite of the considerable studies, the origin of the flexible editing windows and genome-wide off-target of CBEs are not completely understood.

It has been well known that excessive dosage of Cas9 and sgRNA can lead to elevated off-target mutations[22]. This finding led to the widespread use of directly delivered Cas9-guide RNA (gRNA) ribonucleoproteins (RNPs), which can improve the genome editing specificity by limiting the exposure of genome to gene-editing agents[23]. Similarly, delivery of BE-gRNA RNP can increase the base editing specificity compared with transfection of BE-coding plasmids[24]. Besides the use of gene-editing agents with short intracellular half time[23], molecular switches provide compelling opportunities to improve the specificity of CRISPR-Cas via the temporal control of its cellular activity. Indeed, a variety of CRISPR-Cas inhibitors have been developed, including naturally occurring anti-CRISPR (Acr) proteins[25–27], synthetic oligonucleotides[28] and small molecules[29]. These inhibitors can improve the genome editing specificity of Cas9 RNPs[30] and modulate the cellular activity of BEs[29].

In a previous study, we have identified bacteriophage-derived peptides G8P_{PD} as inhibitors to *Streptococcus pyogenes* Cas9 (SpCas9)[31]. Herein, we hypothesized that the broad targeting window of CBEs is attributed, at least in part, to the excess intracellular dosage of base editing agents that could be alleviated by timed delivery of G8P_{PD}. Indeed, we found that overexpression of the peptide inhibitors of SpCas9 in human cells could direct BE3 and A3A CBEs to more focused editing windows and facilitate precision gene correction in a cell model of Marfan syndrome[32] and base editing in mouse embryos.

## Results

**G8P_{PD} peptides inhibit CBE targeting of EGFP reporter.** We have reported in a previous study that the periplasmic domain of the major coat proteins (G8P_{PD}) from M13 and f1 bacteriophages (Fig. 1a) can inhibit the in vitro and cellular activities of SpCas9[31]. Because these peptides suppress SpCas9 activity by disrupting SpCas9-sgRNA binding[31], we hypothesized that they could also act as inhibitors to cytidine base editors (CBEs) the activity of which also rely on the sgRNA-guided DNA binding of the SpCas9 domain. In order to evaluate the inhibitory activities of M13 and f1 G8P_{PD} on CBEs, we constructed an enhanced green fluorescent protein (EGFP) reporter carrying an inactivating mutation Y66C at the chromophore (Supplementary Fig. 1a). An sgRNA was designed to target the Y66C mutation site for introducing C-to-T conversion by CBE (Supplementary Fig. 1b). This mutation can correct EGFP-Y66C to wild-type genotype, yielding fluorescent cells that can be readily detected by flow cytometry. Inhibition of A3A CBE by G8P_{PD} peptides will lead to reduced EGFP fluorescence (Supplementary Fig. 1a).

Our previous study has shown that G8P_{PD} peptides bind to sgRNA-free SpCas9 (apo-Cas9) and that maximum SpCas9-inhibiting activity of G8P_{PD} can be achieved by pre-incubating cells with overexpressed G8P_{PD}[31]. Hence, in the present study we transfected G8P_{PD} plasmids at 24 h prior to the transfection of CBE[17], sgRNA and EGFP-Y66C plasmids (Supplementary Fig. 1c). An inactive G8P_{PD} mutant (Mut2)[31] (Fig. 1a) was included to control for the cell stress induced by serial transfections and non-specific effects of peptide expression on base editing (Fig. 1b). It was found that A3A CBE targeting of EGFP-Y66C reporter plasmid could efficiently recover EGFP fluorescence (Supplementary Fig. 1d). Pre-incubation of cells with M13 or f1 G8P_{PD} overexpression plasmid reduced more than 50% of GFP activation compared with A3A CBE only or Mut2 G8P_{PD} groups (Fig. 1b). These results supported our hypothesis that M13 and f1 G8P_{PD} could also function as inhibitors to BE. Next-generation sequencing (NGS) analyses of the PCR amplification product of A3A CBE-targeted site revealed evident C-to-T mutations in EGFP reporter (Fig. 1c). M13 and f1 G8P_{PD}, but not the inactive G8P_{PD} mutant, could efficiently inhibit CBE-induced C-to-T mutations in EGFP reporter gene (Fig. 1c). It appeared that f1 G8P_{PD} has significantly higher inhibitory activity than M13 G8P_{PD} (Fig. 1b,c). Thus, subsequent studies are performed with f1 G8P_{PD}.

**G8P_{PD} inhibits CBE targeting at endogenous genomic sites in human cells.** To examine the inhibitory effects of G8P_{PD} on CBE at endogenous sites, we designed sgRNA targeted to human *EPPK1*, *GATAD2A*, *DNMT3B*, and *DDX53* genes respectively for A3A CBE (Fig. 1d). Similar to the procedure in EGFP reporter assay, f1 G8P_{PD}-encoding plasmid was transfected into HEK293T cells at 24 h prior to A3A CBE and sgRNA transfection (Fig. 1e). The sgRNA-encoding and f1 G8P_{PD}-encoding plasmids carried GFP and mCherry reporter genes respectively. A G8P_{PD}-free pcDNA3.1 vector was included to control for cell stress induced by serial transfections. In order to enrich cells with transfected plasmids, we used flow cytometry to enrich GFP and mCherry-dual positive cells (Supplementary Fig. 2) for analyses of CBE-induced genomic mutations.

NGS analyses of sorted HEK293T cells revealed efficient base editing by A3A CBE at the four selected genomic sites (Supplementary Fig. 3a). The presence of f1 G8P_{PD} reduced CBE-mediated C-to-T conversion by various degree across different positions and genomic loci (Supplementary Fig. 3a). Statistical analyses of the editing efficiency at all examined positions revealed significant inhibitory activity of G8P_{PD} toward A3A CBE (Fig. 1f). Comparing

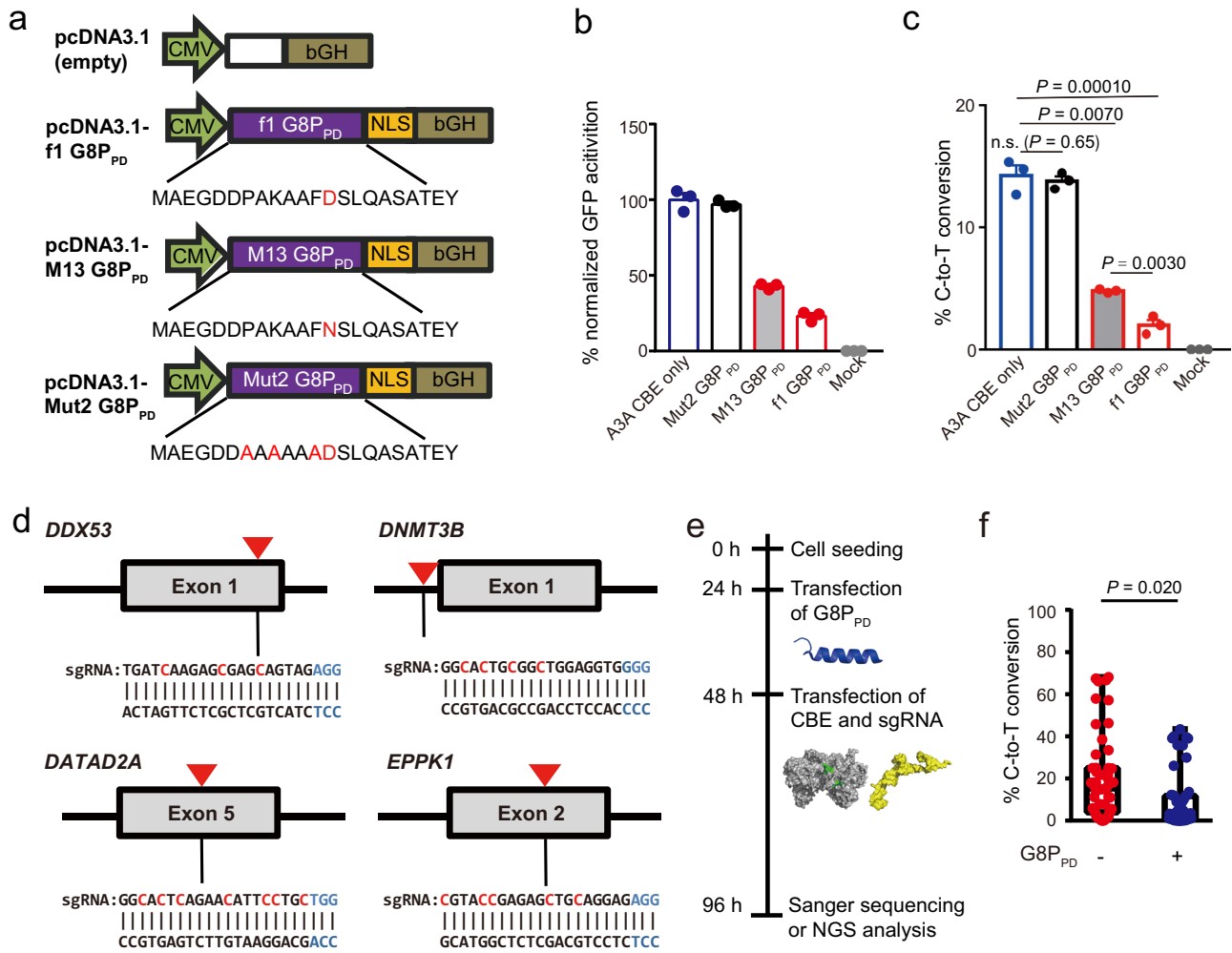

**Fig. 1 G8PPD peptides inhibit CBE targeting of EGFP reporter and endogenous genomic sites in human cells. a** The design of G8P_{PD} plasmids for mammalian expression. **b–c** Analysis of the effects of G8P_{PD} on A3A CBE using EGFP reporter cells. **b** Normalized flow cytometry results. **c** Next-generation sequencing analysis of C-to-T conversion. For **b–c**, the data are shown as mean ± standard error of mean (SEM) ($n = 3$ biologically independent replicates). **d** Design of sgRNA targeted to *DDX53, DNMT3B, GATAD2A* and *EPPK1* genomic loci. The PAM sequence and cytidines are highlighted in blue and red, respectively. Red arrows denote the positions of selected target sites. **e** Flow chart showing experimental procedures. The cartoons are blue for G8P_{PD}, grey for CBE and yellow for sgRNA, respectively. **f** Analysis of the frequency of C-to-T conversion at all editing positions in the absence and presence of G8P_{PD}. Each sgRNA contains 3 biologically independent replicates. The center line in each box indicates the median. The lower and upper bounds of each box represent the first quartile (25%) and the third quartile (75%), respectively. The bottom and top of whiskers denote the minimum and maximum, respectively. For **c** and **f**, significant difference is determined using two-tailed Student's *t* test. n.s., not significant. Source data are provided as a Source Data file.

different target sites or sgRNA, we found that the averaged inhibition rates of each target site had no significant difference (Supplementary Fig. 3b), suggesting that the CBE-inhibiting activity of G8P_{PD} was not dependent on genomic loci or sgRNA.

To investigate whether the activity of G8P_{PD} was cell-type dependent, we examined the performance of G8P_{PD} in U-2 OS cells. It was found that G8P_{PD} reduced the editing efficiency of A3A CBE by various degree across different genomic sites and editing positions (Supplementary Fig. 4a) in U-2 OS. Similar to the results in HEK293T cells, the inhibitory activity of G8P_{PD} in U-2 OS cells was not dependent on genomic loci or sgRNA (Supplementary Fig. 4b). Collectively, these results suggested that G8P_{PD} could inhibit the editing activity of A3A CBE at different genomic sites across different cell types.

**G8P_{PD} differentially inhibits the on-target and out-of-window editing of A3A CBE.** We next analyzed the inhibitory activity of

G8P_{PD} at different editing positions within each target site. It was found that the inhibitory effects of G8P_{PD} displayed notable variations along the 20-bp targeting site for each sgRNA. This result was consistently observed in HEK293T and U-2 OS cells (Fig. 2a). This observation prompted us to compare the effects of G8P_{PD} on the editing positions 4 to 8, which are conventionally deemed as the on-target editing window of CBEs, and on the out-of-window editing positions 1 to 3 and 9 to 20. It was found that G8P_{PD} significantly inhibited both the on-target and out-of-window editing of A3A CBE in HEK293T cells, with the latter being reduced to minimum level (Fig. 2b). Interestingly, G8P_{PD} had minor or little inhibition toward the on-target editing of A3A CBE but significantly inhibited the out-of-window editing (Fig. 2b).

The differential effects of G8P_{PD} at the on-target and out-of-window editing positions were more evident when the inhibition rates were plotted along the 20-bp targeting site (Fig. 2c). Notably, G8P_{PD} exhibited lower inhibition of A3A CBE activity at

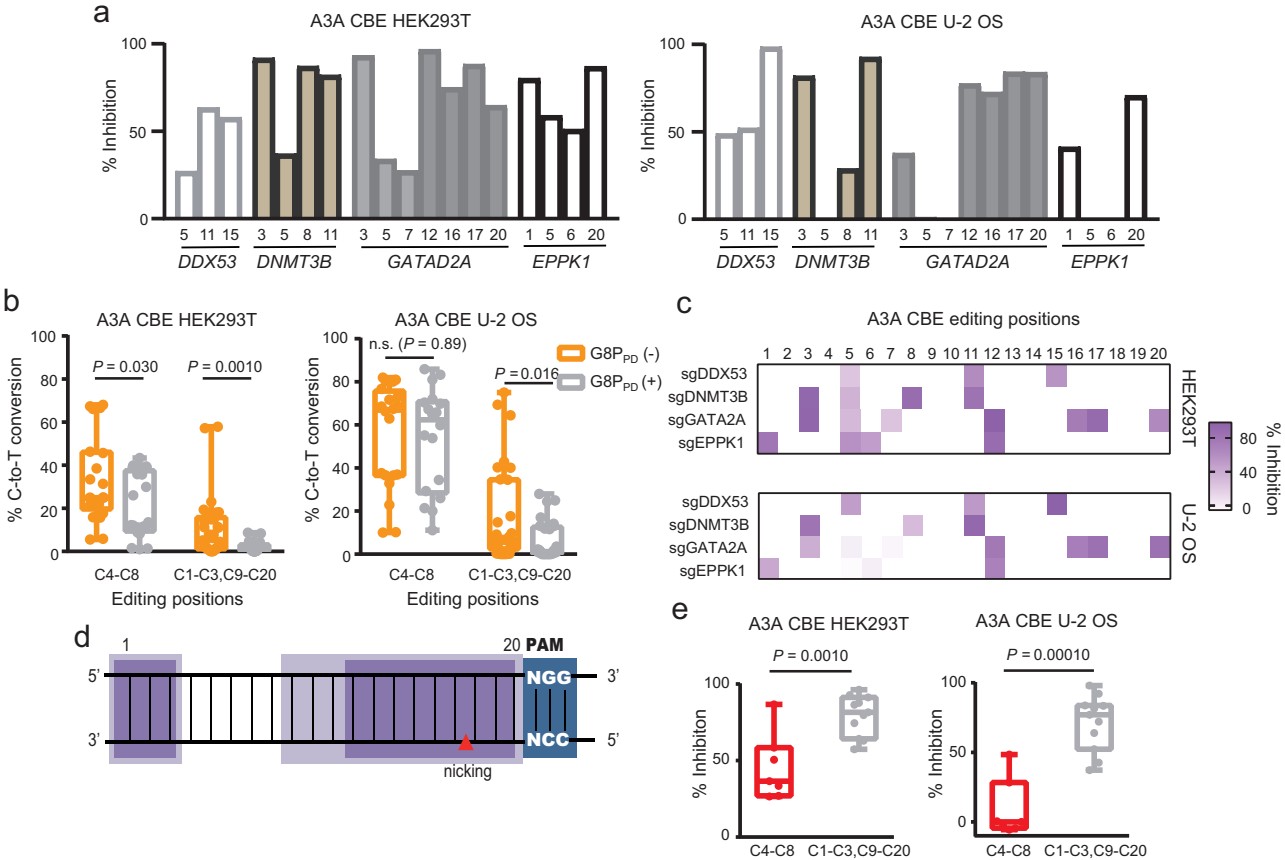

**Fig. 2 The inhibitory activities of f1 G8PPD toward A3A CBE at the endogenous genomic sites in human cell lines. a** The inhibitory activity of f1 G8P$_{PD}$ toward A3A CBE in HEK293T and U-2 OS cells. **b** Position-dependent inhibitory activity of G8P$_{PD}$ at on-target and out-of-window editing positions of A3A CBE. **c** Heat map showing position-dependent inhibition of A3A CBE by G8P$_{PD}$. **d** Schematic presentation of the canonical out-of-window position of CBEs (light purple) and the hot spots of G8P$_{PD}$ inhibition (dark purple). The data are derived from Fig. 2c. PAM sequences are highlighted in blue. **e** Comparison of the inhibition rates of G8P$_{PD}$ at the on-target and out-of-window editing positions of A3A CBE. For **a–e**, *DDX53* and *EPPK1* groups contain 2 biologically independent replicates and *DNMT3B* and *GATAD2A* groups contain 3 biologically replicates. For **b** and **e**, the center line in each box indicates the median. The lower and upper bounds of each box represent the first quartile (25%) and the third quartile (75%), respectively. The bottom and top of whiskers denote the minimum and maximum, respectively. Significant difference is determined using two-tailed Student's *t* test. n.s., not significant. Source data are provided as a Source Data file.

positions 4 to 8, the canonical on-target positions of CBEs (Fig. 2b), than at out-of-window positions. The hot spots of G8P$_{PD}$-mediated inhibition had significant overlap with the canonical out-of-window editing positions of CBEs (Fig. 2d). Comparative analyses of the inhibition of C-to-T conversion showed that G8P$_{PD}$ exhibited 2- and 10-fold selectivity of inhibition toward out-of-window positions over on-target positions in HEK293T and U-2 OS cells respectively (Fig. 2e).

**G8P$_{PD}$ differentially inhibits the on-target and out-of-window editing of BE3 CBE and ABE7.10.** Because G8P$_{PD}$ acts as a SpCas9 inhibitor, we envisioned that G8P$_{PD}$ could inhibit different CBEs carrying SpCas9 module as the DNA-binding domain. Hence, we sought to examine the effects of G8P$_{PD}$ on BE3 CBE and ABE7.10 that contain deaminase domains different from that in A3A CBE. We found that f1 G8P$_{PD}$ could suppress BE3 CBE-induced C-to-T conversion though the inhibitory effects appeared to be less prominent than that with A3A CBE (Supplementary Fig. 5a). Similar to the results with A3A CBE, the inhibitory effects of f1 G8P$_{PD}$ displayed varied inhibition rates at different editing positions of BE3 CBE (Fig. 3a), with the on-target positions 4 to 8 exhibiting lower inhibition compared to the out-of-window positions (Fig. 3b). It was observed that G8P$_{PD}$

had 50-fold selectivity of inhibition toward the out-of-window positions over on-target positions (Fig. 3c).

We next examined the effects of G8P$_{PD}$ on ABE7.10. We designed four sgRNAs targeting to different genomic sites (Fig. 3d). Similar to the results with A3A and BE3 CBEs, f1 G8P$_{PD}$ inhibited the A-to-G conversion activity of ABE7.10 in a position-dependent manner with the on-target positions 4 to 7[13] exhibiting lower inhibition rates than the out-of-window positions (Fig. 3e). f1 G8P$_{PD}$ showed 20-fold selectivity of inhibition toward the out-of-window positions over on-target positions of ABE7.10 (Fig. 3f). These results collectively demonstrated that G8P$_{PD}$ could preferentially inhibit the editing activities of CBE and ABE at out-of-window positions, suggesting the potential application of G8P$_{PD}$ as an agent to improve the targeting scope of BEs. In addition, we examined the effects of G8P$_{PD}$ on A3A CBE editing at the ABE7.10 targeting site *VISTA* and consistent inhibitory activity of G8P$_{PD}$ was observed at both the on-target and out-of-window targeting sites of A3A CBE.

As previous studies suggested that timed delivery of Acrs could improve the genome-editing specificity of SpCas9[30], we next sought to investigate the effects of AcrIIA4 on modulating CBE and ABE activities. HEK293T cells were co-transfected with AcrIIA4 plasmid and sgRNA- and BE-coding plasmids (Supplementary Fig. 6a) and the inhibitory activity of AcrIIA4 was

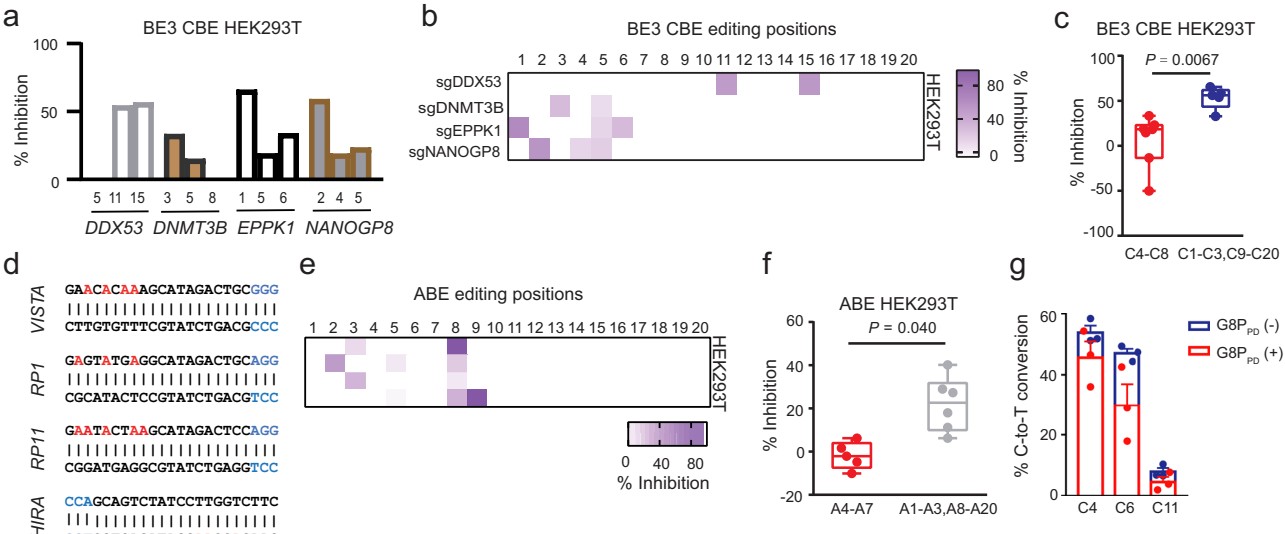

**Fig. 3 The inhibitory activities of f1 G8PPD toward BE3 CBE and ABE7.10 at the endogenous genomic sites in human cell lines. a** The inhibitory activity of f1 G8P$_{PD}$ toward BE3 CBE in HEK293T cells. **b** Heat map showing position-dependent inhibition of BE3 CBE by G8P$_{PD}$. **c** Comparison of the inhibition rates of G8P$_{PD}$ at the on-target and out-of-window editing positions of BE3 CBE. For **a–c**, *DDX53*, *DNMT3B* and *EPPK1* groups contain 2 biologically independent replicates and *NANOGP8* groups contain 3 biologically replicates. **d** Design of sgRNA for ABE7.10 targeted genomic loci. The PAM sequence and adenine positions are highlighted in blue and red, respectively. **e** Heat map showing the position-dependent inhibition of ABE7.10 by G8P$_{PD}$. **f** Comparison of the inhibition rates of G8P$_{PD}$ at the on-target and out-of-window editing positions of ABE7.10. For **c** and **f**, the center line in each box indicates the median. The lower and upper bounds of each box represent the first quartile (25%) and the third quartile (75%), respectively. The bottom and top of whiskers denote the minimum and maximum, respectively. Significant difference is determined using two-tailed Student's *t* test. n.s., not significant. **g** Analysis of the effects of G8P$_{PD}$ on A3A CBE targeting at the ABE7.10 targeting site *VISTA*. The data are shown as mean ± SEM. For **e–g**, each sgRNA contains 3 biologically independent replicates. Source data are provided as a Source Data file.

examined. It was found that AcrIIA4 suppressed the on-target and out-of-window activities of ABE and CBE to undetectable levels (Supplementary Fig. 6b) and that the strong inhibitory activities were observed across all positions within each sgRNA or genomic site (Supplementary Fig. 6c-d).

In addition, previous studies have revealed that the dosage of transfected plasmids can affect the activity and specificity of SpCas9-based genome editing[22] and base editing[33] tools. Therefore, in the present study we sought to investigate the cooperative effects of plasmid dosing and G8P$_{PD}$ on the editing activity of BEs. It was found that under fixed sgRNA plasmid concentration, higher dosages of transfected A3A CBE plasmid could result in increased editing efficiency at *DDX53* and *GATA2A* sites (Supplementary Fig. 7). Importantly, administration of G8P$_{PD}$ notably reduced the editing events at out-of-window positions while maintaining the majority of the on-target activity of A3A CBE (Supplementary Fig. 7).

**G8P$_{PD}$ mediates precision correction of pathogenic *FBN1* mutation by CBEs.** To explore the potential therapeutic application of G8P$_{PD}$, we assessed the effects of G8P$_{PD}$ on the editing activities of A3A and BE3 CBEs in a cell-based disease model of Marfan syndrome[32]. In this cell model, a T7498C mutation was introduced into the *FBN1* gene of HEK293T cells to model the pathogenic amino acid mutation C2500R[32]. To enable CBE-mediated gene correction, we first designed sgRNA for A3A and BE3 CBEs to convert the pathogenic mutation T7498C to wild type (Fig. 4a). HEK293T-*FBN1*$^{T7498C}$ cells were transfected with A3A and BE3 CBEs respectively in the absence or presence of G8P$_{PD}$. Sanger sequencing of the PCR amplification products of *FBN1* gene from edited cells revealed efficient CBE-induced C-to-T conversion (Fig. 4b). NGS analysis showed that A3A and BE3 CBEs induced C-to-T conversion with varied frequencies across the 20-bp target site (Fig. 4c). Similar to the previous results,

G8P$_{PD}$ displayed remarkably higher inhibition rates at out-of-window positions than at on-target positions (Fig. 4d), with 36- and 3-fold selectivity for A3A and BE3 CBEs respectively (Fig. 4e). Most importantly, G8P$_{PD}$ reduced the frequency and the number of genotypes of incorrectly edited alleles of A3A and BE3 CBEs (Supplementary Figs. 8-9) and increased perfectly edited alleles of A3A CBE from less than 4% to more than 38% of the whole population, the latter of which corresponds to more than 50% of the total edited alleles (Fig. 4f). This 9-fold improvement demonstrated the feasibility of using G8P$_{PD}$ as an agent for precision gene correction in Marfan syndrome model.

We also investigated the effects of G8P$_{PD}$ on high-fidelity CBE variants for correction of *FBN1*$^{T7498C}$ mutation. It was found that G8P$_{PD}$ could modestly yet significantly inhibit the out-of-window editing activity of hA3A-eBE-Y130F at C1 position without affecting its on-targeting activity at C4 position (Fig. 4g). For BE3-R33A that exhibited activity only at C4 position, G8P$_{PD}$ did not affect its on-target editing. These results together suggested that peptide inhibitors of SpCas9 such as G8P$_{PD}$ are compatible with and may even improve high-fidelity BE variants for precision gene correction.

**Co-injection of G8P$_{PD}$ mRNA mediates precision base editing in mouse embryos.** In order to avoid serial transfection of G8P$_{PD}$ for practical applications, we investigated the effects of co-transfected G8P$_{PD}$-encoding plasmid on base editing. It was found that pre-transfection, but not co-transfection, of G8P$_{PD}$ plasmid could reduce the out-of-window editing of A3A CBE (Supplementary Fig. 10a-b). This result was consistent with the previous finding that timed delivery of G8P$_{PD}$ was important for achieving different effects on SpCas9-based genome editing tool[31].

We next examined the effects of co-injected G8P$_{PD}$-encoding mRNA on A3A CBE-mediated base editing in mouse embryos. Expression of G8P$_{PD}$ from mRNA could bypass the transcription

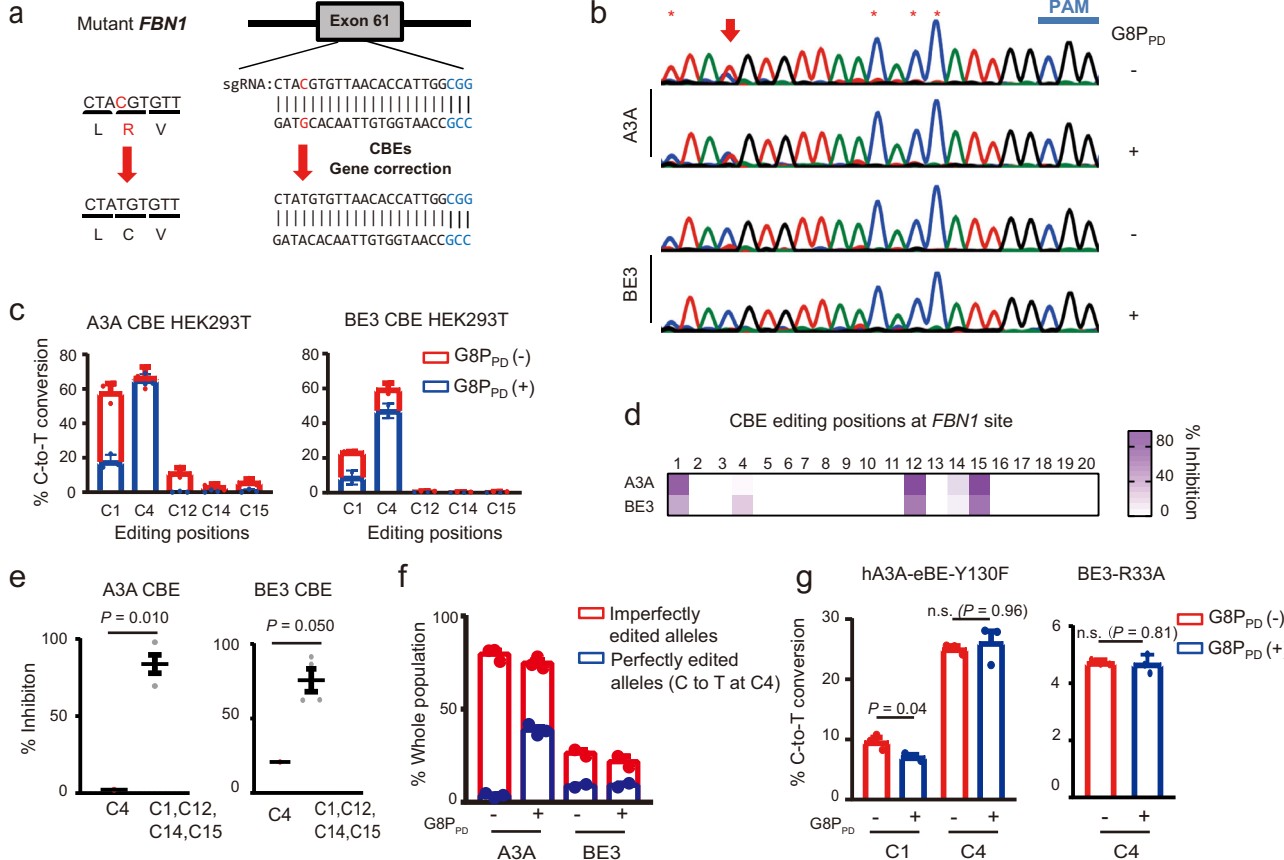

**Fig. 4 G8PPD improves the targeting window of CBEs for correction of FBN1T7498C mutation. a** Schematic illustration of sgRNA design and the strategy for CBE-based gene correction. **b** Sanger sequencing of the positive strand of *FBN1* illustrating CBE-induced C-to-T mutations. Arrows and asterisks denote on-target and out-of-window cytidine positions for base editing. **c** Frequency of A3A and BE3 CBE-mediated C-to-T conversion in the absence and presence of f1 G8P$_{PD}$. The data are shown as mean ± SEM. The A3A and BE3 groups contain 2 and 3 biologically independent replicates respectively. **d** Heat map showing the position-dependent inhibition of A3A and BE3 CBEs by G8P$_{PD}$. **e** Comparison of the inhibition rates of G8P$_{PD}$ at the on-target and out-of-window editing positions of A3A and BE3 CBEs at *FBN1* site. Significant difference is determined using Student's *t* test. **f** Analyses of perfectly edited *FBN1* alleles in the absence and presence of G8P$_{PD}$. For **e**–**f**, the data are shown as mean ± SEM and the A3A and BE3 groups contain 2 and 3 biologically independent replicates respectively. **g** The effects of f1 G8P$_{PD}$ on hA3A-eBE-Y130F and BE3-R33A CBEs for correction of *FBN1*$^{T7498C}$ mutation. The data are shown as mean ± SEM (*n* = 3 biologically independent replicates). For **e** and **g**, significant difference is determined using two-tailed Student's *t* test. n.s., not significant. Source data are provided as a Source Data file.

process and resemble pre-transfection of G8P$_{PD}$. A previously described sgRNA targeting to mouse *Tyr* gene[34] was constructed for introduction of a stop codon (Fig. 5a). G8P$_{PD}$-encoding mRNA was in vitro transcribed and then co-injected into one-cell stage mouse embryos with sgRNA and A3A CBE-coding plasmids (Fig. 5b). Compared with the mock group containing pcDNA, G8P$_{PD}$ mRNA did not compromise blastocyst development or genomic amplification efficiency (Fig. 5c). NGS analysis of successfully amplified gene alleles showed that the use of G8P$_{PD}$ could facilitate the generation of blastocysts carrying perfectly edited gene alleles despite of the slightly lower overall editing efficiency compared to mock group (Fig. 5d and Supplementary Fig. 11).

To expand the generality and applicability of our finding, we performed microinjection experiments using sgRNA and CBE-coding mRNAs. In these experiments, we used chemically synthesized G8P$_{PD}$ peptide to ensure the presence of sufficient G8P$_{PD}$ for modulating the activity of transcribed A3A and BE3 CBE. In consistency with the above observations, G8P$_{PD}$ peptide improved the efficiency of generating perfectly edited blastocysts using A3A or BE3 CBE-coding mRNA (Fig. 5d and Supplementary Fig. 12-13). Thus, G8P$_{PD}$ may function as an efficient tool for generation of genetically modified animals.

## Discussion

It has well documented that CRISPR-Cas systems are associated with off-target cleavage[35], chromosomal rearrangement[36] and genotoxicity[37]. These adverse effects may result from uncontrolled expression of Cas proteins[22,38,39]. Unfortunately, most therapeutically relevant applications of CRISPR-Cas involve constitutively active Cas nucleases that raises major safety concern[40]. With the increasing number of therapeutic applications of CRISPR-Cas, the ability to manipulate the activity of CRISPR-Cas in human cells has become urgently needed. One feasible approach is to develop CRISPR-Cas off-switches to control the intracellular activity of Cas proteins. Thus far, synthetic oligonucleotides[28], small-molecule inhibitors[29] and bacteriophage-derived anti-CRISPR proteins (Acrs)[31] have been developed to enable the temporal control of CRISPR-Cas activity. It has been shown that Acrs can improve the targeting specificity of SpCas9 at on-target sites over off-target sites[30]. This improvement was achieved via precisely controlled, timed delivery of Acrs along with Cas9-sgRNA ribonucleoproteins (RNPs)[30]. Yet, the effects of Acrs on base editors have not been explored. Notably, small-molecule inhibitors of SpCas9 have been employed as inhibitors to Cas9-based transcription factors and base editors[29].

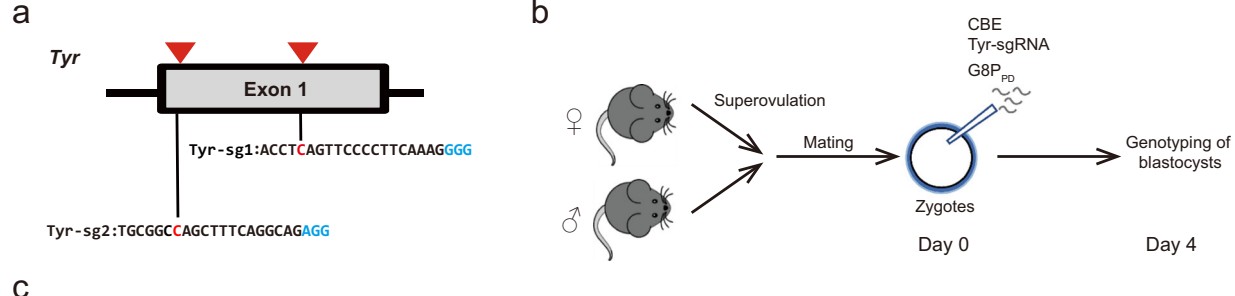

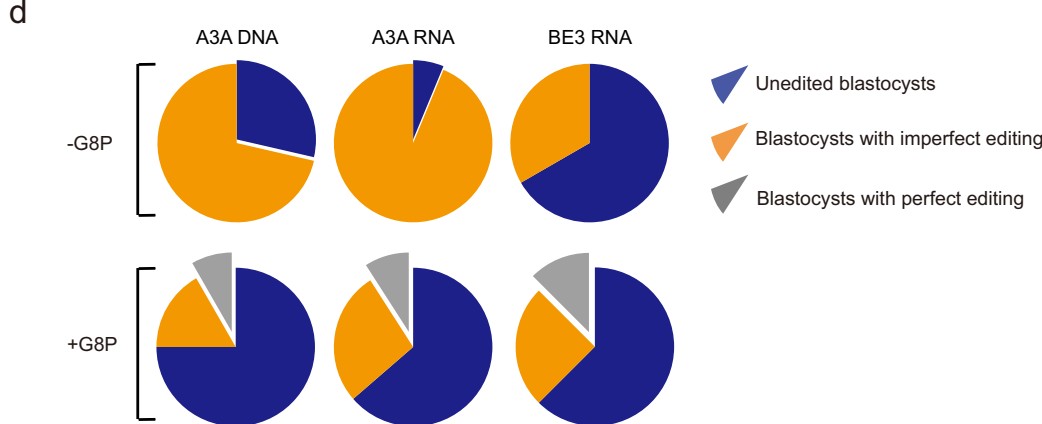

| Groups | | Injected embryos | Harvested blastocysts | Successfully amplified blastocysts | Edited blastoysts | Blastocysts with perfect editing |
|---|---|---|---|---|---|---|
| Tyr-sg1 + A3A + pcDNA | Microinjection DNA | 47 | 10 (21.28%) | 7 | 5/7 (71.43%) | 0/5(0) |
| Tyr-sg1 + A3A + G8P$_{PD}$ mRNA | Microinjection DNA | 34 | 16 (47.05%) | 12 | 3/12 (25%) | 1/3(33.33%) |
| Tyr-sg1 + A3A | Microinjection RNA | 38 | 32 (84.21%) | 32 | 30/32 (93.75%) | 0/30(0) |
| Tyr-sg1 + A3A + G8P$_{PD}$ peptide | Microinjection RNA | 20 | 12 (60%) | 11 | 4/11 (36.36%) | 1/3(33.33%) |
| Tyr-sg2 + BE3 | Microinjection RNA | 20 | 12 (60%) | 6 | 2/6 (33.33%) | 0/2 (0) |
| Tyr-sg2 + BE3 + G8P$_{PD}$ peptide | Microinjection RNA | 24 | 16 (66.67%) | 8 | 3/8 (37.5%) | 1/3 (33.33%) |

**Fig. 5 Co-injection of G8P$_{PD}$ mRNA or peptide facilitates precision base editing of mouse embryos using CBE-coding plasmids or mRNAs. a** Design of sgRNA targeting to mouse *Tyr* gene for introduction of stop codon. The PAM sequence and targeted cytidine are shown in blue and red, respectively. **b** Schematic presentation showing experimental procedures. **c** The outcome of base editing in mouse embryos in the absence and presence of G8P$_{PD}$. **d** Analysis of CBE-induced mutations in amplified gene alleles of blastocysts. Source data are provided as a Source Data file.

In our previous study, we discovered that the in vitro and cellular activity of SpCas9 can be inhibited by the periplasmic domain of bacteriophage major coat proteins (G8P$_{PD}$)[31]. These G8P$_{PD}$ peptides disrupt Cas9-sgRNA assembly by binding to the PAM interacting (PI) domain of SpCas9[31]. It was found that time-controlled delivery of overexpressed G8P$_{PD}$ could improve the specificity of SpCas9 in human cells[31]. In the present study, we hypothesized that the out-of-window base editing of CBE may be attributed to excess CBE fusion proteins. Hence, we sought to explore whether timed delivery of SpCas9-inactivating G8P$_{PD}$ could inhibit and improve the activities of CBE and ABE, in a rationale similar to that with the genome editing of SpCas9[31]. Our results have demonstrated that G8P$_{PD}$ can preferentially inhibit the out-of-window editing of A3A and BE3 CBEs and ABE7.10, leading to more focused targeting window. To the best of our knowledge, our study represents the first peptide inhibitors that can improve the targeting scope of BEs. Unlike G8P$_{PD}$ that preferentially inhibited out-of-window over on-target editing, AcrIIA4 abolished the activities of CBE and ABE at both on-

target and out-of-window sites without any selectivity. These results support the notion that weak inhibitors, rather than strong inhibitors, are optimal agents for modulating the specificity of genome-editing tools. Nevertheless, it is possible that the editing activity of BEs may be improved by limiting the exposure time of SpCas9 to Acrs where Acrs are delivered at later time points during transfection.

It has been reported that the targeting specificity of CBEs and ABEs can be improved by incorporating genetic mutations into deaminase domain[20,21]. These BE variants are extremely useful when precision base editing is needed for therapeutic applications. In the present study, we have shown that G8P$_{PD}$ has minor yet significant inhibition at the out-of-window editing position of hA3A-eBE-Y130F CBE. Most importantly, G8P$_{PD}$ dose not affect the on-target editing activities of hA3A-eBE-Y130F and BE3-R33A CBEs. These results have established the feasibility of G8P$_{PD}$ not only as an alternative approach to improving the targeting specificity of BEs but also as an additive agent to combine with BE variants. In further studies, it would be

interesting to evaluate the inhibitory activities of G8P$_{PD}$ toward dual adenine and cytidine base editors (A&C-BEs)[41] and prime editors[42]. These studies can advance our understanding of the capability of G8P$_{PD}$ serving as general modulators of CRISPR-Cas genome engineering tools.

As a proof of concept, most of the described applications of G8P$_{PD}$ in the present study was carried out through serial transfection. Although co-transfection of G8P$_{PD}$ plasmid with sgRNA and BE-coding plasmids did not improve the targeting specificity of A3A CBE, co-injection of G8P$_{PD}$ mRNA with sgRNA and BE-coding plasmids into mouse embryos could facilitate precision base editing for generation of blastocysts carrying perfectly edited genes. Similarly, G8P$_{PD}$ peptide could facilitate precision base editing in mouse embryos using A3A or BE3 CBE-coding mRNA. It must be noted, however, that further studies with larger-scale analysis of the effects of G8P$_{PD}$ on embryonic base editing would be important to reveal the full potential of G8P$_{PD}$. In addition, it may be also feasible to genetically fuse G8P$_{PD}$-coding sequence with BEs to facilitate the application of G8P$_{PD}$. However, due to the specific mechanism of action of G8P$_{PD}$, which allosterically inhibits SpCas9[31], the structural organization of G8P$_{PD}$-BE fusion protein may need to be carefully investigated.

Although we only examined f1 and M13 G8P$_{PD}$ for inhibition of CBE activity in the current study, there might be other G8P$_{PD}$ from inoviridae phages[31] that can function as CBE off-switches. Comparative analyses can be performed on these different G8P$_{PD}$ peptides to determine the sequence-activity relationship. This information would facilitate the design and development of next-generation peptide off-switches of CBEs. Moreover, it can be also important to characterize the mechanism of action of G8P$_{PD}$ on CBE inhibition. Unlike most Acrs that inhibit Cas activities by disrupting DNA or RNA binding, G8P$_{PD}$ functions as an allosteric inhibitor of SpCas9[31]. This unique mechanism may be critical for G8P$_{PD}$ to selectively modulate BE activities at on-target and out-of-window positions. The in-depth mechanism studies could facilitate the development of next-generation CRISPR modulators for precision gene correction.

## Methods

**Animal ethics statement.** The use and care of animals were complied with the guidelines of the Institutional Animal Care and Use Committee (IACUC) of GemPharmatech Co., Ltd, Nanjing, Jiangsu, China. Mice were maintained in a SPF (specific pathogen-free) facility under a 12 h dark-light cycle.

**Plasmid construction.** sgRNA was cloned into pGL3-sgRNA expression vector carrying a U6 promoter and an EGFP reporter gene (Addgene, #107721). CBE-targeted genomic sites are indicated in Supplementary Table 1. The sequences of sgRNA-encoding oligonucleotides were listed in Supplementary Table 2. Human codon-optimized DNA sequences encoding M13, f1 G8P$_{PD}$ and Mut2 G8P$_{PD}$ were cloned into the *BamH* I/*Xba* I sites of pcDNA3.1(+) by plasmid recombination kit Clone Express (Vazyme Biotech, Nanjing, China). These G8P$_{PD}$ peptides carry a C-terminal SV40 nuclear localization signal (NLS) for co-localization with Cas9 proteins. G8P$_{PD}$ peptides were cloned into plv-EF1α-mCherry plasmid harboring mCherry fluorescent protein marker. For construction of EGFP-Y66C reporter plasmid, Y66C mutation was introduced by quickchange PCR. Mut2 G8P carrying a C-terminal SV40 nuclear localization signal (NLS) was cloned into A3A CBE expression vector.

**Cell culture and transfection.** HEK293T and U-2 OS cells were obtained from ATCC and maintained in Dulbecco's Modified Eagle Medium (DMEM) (Cat. No. SH30243, Hyclone, Logan, USA) supplemented with 10% fetal bovine serum (FBS) and 1% penicillin-streptomycin at 37 °C with 5% CO$_2$. Transfection was performed using lipofectamine 2000 Reagent (Cat. No. 11668019, ThermoFisher Scientific, Waltham, USA) according to manufacturer's instructions. HEK293T and U-2 OS cells were seeded on to poly-D-lysine (Cat. No. A-003-E, Sigma, St. Louis, USA) pre-coated 24-well plates at 24 h prior to transfection.

For EGFP reporter assay, 1 µg G8P$_{PD}$-encoding plasmid was transfected into HEK293T cells. After 24 h, 0.25 µg sgRNA plasmid, 0.5 µg A3A CBE plasmid and 0.05 µg the EGFP-Y66C reporter plasmid were co-transfected into G8P$_{PD}$-expressing cells. At 48 h after the second transfection, cells were harvested and

analyzed by flow cytometry. The efficiency of GFP activation was calculated based on sorted cells. To calculate the C-to-T conversion in EGFP gene, the cells were harvested without flow cytometry enrichment and then analyzed by NGS. For base editing at endogenous genes, unless noted otherwise, cells of approximately 70% confluency were transfected with 1 µg G8P$_{PD}$ plasmid that carries an mCherry selection marker at 12 h after seeding. At 24 h after G8P$_{PD}$ transfection, 0.25 µg sgRNA plasmid that carries a GFP selection marker and 0.5 µg CBE plasmid were transfected in G8P$_{PD}$-expressing cells. At 48 h after CBE and sgRNA plasmids transfection, cells were harvested and analyzed by Beckman Coulter CytFLEX (Beckman Coulter, Brea, USA) or sorted by BD FACSAria III flow cytometry (BD Biosciences, New York, USA). FlowJo VX or FlowJo 10 was used to analyze flow cytometry data. At least 2,000 mCherry and GFP dual positive cells were collected for subsequent analyses. Analyses of the Sanger sequencing results of the PCR product of edited EGFP reporter were performed using EditR[43] and analyzed by NGS.

For AcrIIA4 co-transfection experiments, HEK293T cells were transfected with 0.5 µg of CBE plasmid and 0.25 µg sgRNA-expressing and 1 µg of AcrIIA4-expressing plasmids that carry GFP and mCherry reporters, respectively. At 72 h after transfection, mCherry and GFP dual positive cells were collected for subsequent analyses.

**In vitro transcription.** G8P$_{PD}$ mRNA was transcribed from f1 G8P$_{PD}$-coding PCR products with a 5′ T7 promoter sequence using HiScibe T7 ARCA mRNA kit (NEB). The transcription was performed at 37 °C overnight and then purified by phenol: chloroform extraction, followed by ethanol precipitation. To generate BE3 and A3A CBE mRNA, BE3 and A3A-coding plasmids were linearized by AgeI (NEB, R3552L) and then used as templates for in vitro transcription using T7 ULTRA (Ambion, AM1345). BE3 and A3A mRNAs were purified using RNeasy Mini Kit (QIAGEN, 74104). The RNA of *Tyr*-targeting sgRNA was amplified and transcribed from a sgRNA-coding plasmid along with a T7 promoter using MEGA short transcript T7 KIT (Ambion, AM1345). The sgRNAs were purified using MEGA clear Kit (Ambion, AM1908) and recovered by alcohol precipitation.

**Microinjection of mouse one-cell embryos.** Female C57BL/6 mice of 3.5–4 weeks old and male C57BL/6 mice of 3–6 months old were used in our experiments. All mice were housed in the specific pathogen-free (SPF) animal facility at Gem-Pharmatech Co., Ltd in accordance with institutional guidelines under the following conditions: 23 °C ambient temperature, 40-70% humidity, 12 h dark/light cycle and free access to water and rodent chow. Superovulated female C57BL/6 J mice were mated to C57BL/6 J males, and zygotes were collected from the oviducts at E0.5. For DNA injection, 100 ng G8P$_{PD}$ mRNA, 50 ng A3A plasmid and 25 ng sgRNA plasmid were mixed and injected into the cytoplasm of zygotes with well recognized pronuclei. For RNA injection, 0.5 ng G8P$_{PD}$ peptide, 50 ng A3A mRNA and 25 ng sgRNA were mixed and injected into the cytoplasm of zygotes with well recognized pronuclei. Injected zygotes were cultured to blastocysts and then harvested for genomic analysis.

**Extraction and PCR amplification of genomic DNA.** Genomic DNA was extracted using QuickExtract™ DNA Extraction Solution (Lucigen, USA). Genomic PCR was performed using 100 ng genomic DNA, corresponding primers (Supplementary Table 3-4), Phanta Max Super-fidelity DNA Polymerase (Cat. No. P505-d1, Vazyme) or KOD plus (Cat. No. F0934K, Takara, Kyoto, Japan) with a touchdown cycling protocol that contains 30 cycles of 98 °C for 10 s, X °C for 15 s where X decreases from 68 °C to 58 °C with a −1 °C/cycle rate and 68 °C for 60 s.

**NGS analyses of PCR amplicons.** PCR was performed using the NGS primers (Supplementary Table 4). PCR products were purified using Gel Extraction Kit (Cat. No. D2500-02, OMEGA) before construction of NGS libraries. Hiseq3000 SBS&Cluster high-throughput NGS library preparation kit (Cat. No. FC-410-1002, Illumina, San Diego, CA, USA), (VAHTS Universal DNA Library Prep Kit (ND608-01, Vazyme) or TruSeq NanoDNALT Library Prep Kit (Illumina) were used to generate dual-indexed sequence following the manufacturer's protocol. Briefly, more than 50 ng purified PCR fragment was used for direct library preparation. The fragments were treated with End Prep Enzyme Mix (Cat. No. WE0229, Illumina) for end repairing, 5′ phosphorylation and dA-tailing in one reaction, followed by a T-A ligation to add adaptors to both ends. Size selection of adaptor-ligated DNA was then performed using VAHTSTM DNA Clean beads (Cat. No. N411-03, Vazyme) or Beckman AMPure XP beads (Cat. No. A63882, Illumina). Each sample was then amplified by PCR for 8 cycles using P5 and P7 primers (Supplementary Table 4). Both P5 and P7 primers carry the sequences that can anneal with flow cells for bridge PCR. In addition, P7 primer carries a six-base index allowing for multiplexing. The PCR products were cleaned using VAHTSTM DNA Clean beads (Cat. No. N411-03, Vazyme) or Beckman AMPure XP beads (Cat. No. A63882, Illumina), validated using an Agilent 2100 Bioanalyzer (Agilent Technologies, Palo Alto, CA, USA) and quantified by Qubit2.0 Fluorometer (Invitrogen, Carlsbad, CA, USA) or Quant-iT PicoGreen dsDNA Assay Kit (ThermoFisher). Two or three biological replicates were processed by Genewiz (Suzhou, China) or Personalbio (Shanghai, China) using Illumina HiSeq 3000. Sequencing was carried out using a 2 × 150 paired-end (PE) or 2 × 300 paired-end

(PE) configuration. Image analyses and base calling were conducted by the HiSeq Control Software (HCS) + OLB + GAPipeline-1.6 (Illumina) on the HiSeq instrument. Sequencing reads were obtained in the Fastq format. Amplicons with less than 6 M read counts were excluded from the analyses.

To obtain the editing efficiencies, the adapter pair of the pair-end reads were removed using AdapterRemoval version 2.2.2, and pair end read alignments of 11 bp or more bases were combined into a single consensus read. All processed reads were then mapped to the target sequences using the BWA-MEM algorithm (BWA v0.7.16). For each site, the mutation rate was calculated using bam-readcount with parameters -q 20 -b 30. Indels were calculated based on reads containing at least 1 inserted or deleted nucleotide in protospacer. Indel frequency was calculated as the number of indel-containing reads/total mapped reads.

**Statistical analyses**. For cell-based assay, three biological replicates are generally performed and the results are shown as mean ± standard error of the mean (SEM). For NGS analyses, two or three biological replicates were included for each experimental condition. In practice, three biological replicates were performed at first place and samples sent for NGS analysis. If two replicates could be successfully obtained then the data will be collected. Otherwise, the whole experiments will be repeated to keep consistency between parallel groups. The exact number of samples are reflected in each figure by the number of data points. Statistical analyses were performed using GraphPad Prism 8 with two-tailed Student's $t$ test unless otherwise noted.

**Reporting summary**. Further information on research design is available in the Nature Research Reporting Summary linked to this article.

## Data availability
NGS data have been deposited in the NCBI Sequence Read Archive database under the accession code SRP312256 (cell lines; https://www.ncbi.nlm.nih.gov/sra/?term=SRP312256) and PRJNA798574 (embryos; https://www.ncbi.nlm.nih.gov/sra/?term=PRJNA798574). Source data are provided with this paper.

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

## Acknowledgements
We thank the High-Throughput Screening Platform and Biomedical Big Data Platform at Shanghai Institute for Advanced Immunochemical Studies (SIAIS) at ShanghaiTech University for the support of flow cytometry experiments and analyses of NGS data. This work is supported by the National Natural Science Foundation of China (31600686 to J.L.) and ShanghaiTech University Startup Fund (2019F0301-000-01 to J.L.)

## Author contributions
J.L. conceptualized study. J.L., K.J., and Y.-R.C. designed the experiments, S.H. analyzed next-generation sequencing data. K.J. and Y.-R.C. performed the in vivo CBE-inhibiting experiments. Z.L. and P.M. provided critical resources. J.L. and K.J., and Y.-R.C. wrote the manuscript. All authors discussed the results and approved the manuscript.

## Competing interests

The authors declare no competing interests.
