## [Peer Review File · Nature Communications]

Reviewers' Comments:

Reviewer #1:

Remarks to the Author:

General comments

Precise base editing without any byproducts has been sought especially for the use in therapeutic applications. Conventional genome editing which induces DNA double strand breaks inevitably causes stochastic insertions and deletions at target sites, as well as cellular toxicity. Base editing technology that employs deaminase activity for base conversion allows more predictable base change without DNA double strand breaks, while it is often challenging to precisely change at single base resolution, as it has its activity window ranging 3 to 5 bases or even more. Several efforts have been made to narrow down the window and to reduce bystander mutations mostly through engineering of the deaminase domain and its linkage to Cas proteins. In this manuscript, Jia et al addressed this issue by utilizing a bacteriophage-derived peptide, G8PPD, which the same group had previously characterized as a moderate inhibitor for Cas9 by preventing gRNA incorporation. The authors have shown that when the human culture cells are pre-incubated by transfecting a DNA vector overexpressing the peptide, mutation windows of the selected base editors significantly narrowed down with minimum reduction of editing efficiency at the hot spot. Taking advantage of this, they demonstrated efficient and precise correction of the pathogenic mutation in Marfan syndrome model cells. However, their rationale of this effect is not clear and their serial transfection method raises concern of technical issues as well as applicability. The authors "hypothesized that the broad targeting window of CBEs is attributed, at least in part, to the excess intracellular dosage of base editing agents". If that is the case, one can simply reduce dosage of CBE and/or use RNA or RNP instead, as their timed transfection, as proposed, limits practical use such as therapeutics. To claim superiority of their approach, the authors could perform titration of CBE dosage, use RNA or RNP and compare activity window. It would also be more interesting if the authors can demonstrate this in a single transfection or in a more practical setting such as generating a mutant animal or iPS cell. Technically, it is to be clarified if the reduced activity of BEs is really attributed to the specific effect of the peptides or simply due to overexpression of a random protein that affects cellular integrity. To distinguish this, the authors should use the inactive mutant peptide from their previous work as a control experiment. There are also several points that need to be addressed to gain credibility of their study, as described below.

Specific comments

In Fig. 1, although the presence of the peptides reduced the GFP positive cells population, it is difficult to say that this effect is really attributed to these peptides. Appropriate control experiment as mentioned above is needed.

How did the authors perform sequencing for GFP reporter editing? It would be better to explain why the C-to-T conversion rate is much lower compared to the ratio of the corresponding GFP positive cells population in Figure 1c and 1b, respectively.

In Fig. 1 legend, a sentence, "c Flow chart showing experimental procedures.", should be removed. It would be better to explain Fig. 1b and 1c separately, and more clearly.

In page 5 on line 9, what does "lowest inhibition rates" mean?

In Fig. 2a, the position 20 of EPPK1 should be 12.

Which date is Figure 2d based on? As the canonical targeting window of A3A CBE is 2 to 13 (described in Introduction of this manuscript), it would be better to show the inhibition data at position of 2, 4, 9, 10, and 13, or explain if the authors have any additional information.

In Fig. 4b and Supplementary Fig. 1, the Sanger sequence results are too small to see the differences. Supplementary Fig. 1d showed 34.7% double positive cells for A3A CBE, but the G-to-A conversion is not seen with such ratio in the corresponding Sanger sequencing result. Generally,

quantifying mixed populations by Sanger sequencing is difficult and the authors need to describe in detail how they ensure the quantitiveness.

Fig. 4f is not properly labeled.

Sup Fig 5 shows almost no effect on some target. More targets need to be analyzed to assess generality.

In Sup Fig. 6b, y-axis should be A-to-G conversion for ABE

In Sup Fig. 6d, the scale of y-axis is wrong.

Reviewer #2:

Remarks to the Author:

Base editing is a powerful method for inducing a single nucleotide conversion without DNA double-strand break. In this manuscript, Jia et al. presented a method of using a bacteriophage-derived peptide, termed G8Ppd, for fine tuning of the editing window of base editors (BEs) in human cells. They were able to achieve a precise base editing to correct a Marfan syndrome mutation by temporal expression of the phage peptide. I have the following points to improve this manuscript.

1. In Fig. 3d-e, the authors provided the inhibitory activity of G8Ppd for ABEs at five genomic sites. How about the inhibitory activity for CBEs at the same sites?

2. According to the experimental scheme in Fig. 1e, the phage-derived peptide is transfected 24 hours before cells are transfected with BEs. A serial transfection is cumbersome and may not be feasible with certain cell types such as haematopoietic stem cells. I wonder whether the authors have tried co-transfection. Alternatively, the phage-derived peptide may be fused to BEs using a linker or a self-cleaving 2A peptide.

Minor points:

1. In the Fig. 1 legend, the authors should delete "c flow chart showing experimental procedures", which was repeated in Fig. 1d.

Title: Phage Peptides Mediate Precision Base Editing with Focused Targeting Window

Authors: Jia K, *et al.*

MS ID: NCOMMS-20-04881B

Reviewer #1:

General comments:

The authors have shown that when the human culture cells are pre-incubated by transfecting a DNA vector overexpressing the peptide, mutation windows of the selected base editors significantly narrowed down with minimum reduction of editing efficiency at the hot spot. Taking advantage of this, they demonstrated efficient and precise correction of the pathogenic mutation in Marfan syndrome model cells. However, their rationale of this effect is not clear and their serial transfection method raises concern of technical issues as well as applicability.

RESPONSE-We thank the reviewer for raising these concerns. As reported in the previous study (Cui *et al.*, 2020, *Genome Biol*), phage-derived peptide G8P_{PD} binds to the PAM-interacting (PI) domain of SpCas9 and disrupt Cas9-sgRNA assembly. This mechanism of action prompted us to examine the effects of G8P_{PD} on SpCas9-containing base editing tools.

We agree with the reviewer that serial transfection may limit the applicability of our discovery. Therefore, we performed single transfection experiments using *in vitro* transcribed G8P_{PD} mRNA along with BE-containing plasmids on mouse embryos. These results have been added as *Fig. 5*. More details of the experiments are described below.

The authors “hypothesized that the broad targeting window of CBEs is attributed, at least in part, to the excess intracellular dosage of base editing agents”. If that is the case, one can simply reduce dosage of CBE and/or use RNA or RNP instead, as their timed transfection, as proposed, limits practical use such as therapeutics. To claim superiority of their approach, the authors could perform titration of CBE dosage, use RNA or RNP and compare activity window.

RESPONSE-We appreciate that the reviewer raised these interesting questions. It has been known that the dosage of transfected plasmids has notable effects on the intracellular activity and specificity of SpCas9-based genome editing (Hsu *et al.*, 2013, *Nat Biotech*) and base editing (Koblan *et al.*, 2018, *Nat Biotech*). Similarly, we found in the present study that the targeting activity of A3A BE3 across different editing positions was affected by the dosage of transfected plasmids. Importantly, it was found that addition of G8P_{PD} could significantly reduce the out-of-window editing without compromising the on-target activity of A3A BE3 at different plasmid dosages. These new results have been added as the new *Supplementary Fig. 7*.

It would also be more interesting if the authors can demonstrate this in a single transfection or in a more practical setting such as generating a mutant animal or iPS

cell.

RESPONSE-We thank the reviewer for raising this interesting question. We fully agree that single transfection would make G8P_{PD} more appealing for practical applications. We carried out experiments to compare the effects of G8P_{PD} co-transfection and pre-transfection (serial transfection). It was found that pre-transfection, but not co-transfection, of G8P_{PD} could reduce the off-target editing of A3A CBE. The new results are added as the new *Supplementary Fig 10*.

In order to make single transfection a feasible approach for the practical application of G8P_{PD}, we sought to co-transfect *in vitro* transcribed G8P_{PD} mRNA along with BE-coding plasmids. The rationale is that the expression of G8P_{PD} from mRNA can bypass the transcription step and thus can resemble the process of pre-transfection. Using this approach, we showed that co-injection of G8P_{PD} mRNA with BE-encoding plasmids into mouse embryo could generate blastocysts with perfectly edited gene alleles. The new results have been added as *Fig. 5* and *Supplementary Fig 11*. Nevertheless, we realize that these results are limited by the number of processed embryos and that further studies with larger-scale analysis would be important to reveal the full potential of G8P_{PD} on mediating precision embryonic base editing for generation of model animals. We have revised the Discussion section to further address this point.

Technically, it is to be clarified if the reduced activity of BEs is really attributed to the specific effect of the peptides or simply due to overexpression of a random protein that affects cellular integrity. To distinguish this, the authors should use the inactive mutant peptide from their previous work as a control experiment.

RESPONSE-We thank the reviewer for raising this critical point. An inactive G8P_{PD} mutant (Cui *et al.*, 2020, *Genome Biol*) has been included in *Fig. 1b-c* and *Supplementary Fig. 1d* as a control to rule out the non-specific effects of peptide expression on base editing.

Specific comments

In Fig. 1, although the presence of the peptides reduced the GFP positive cells population, it is difficult to say that this effect is really attributed to these peptides. Appropriate control experiment as mentioned above is needed.

RESPONSE-As stated above, we have included an inactive G8P_{PD} mutant (Cui *et al.*, 2020, *Genome Biol*) in *Fig. 1b-c* as a control to exclude non-specific effects.

How did the authors perform sequencing for GFP reporter editing? It would be better to explain why the C-to-T conversion rate is much lower compared to the ratio of the corresponding GFP positive cells population in Figure 1c and 1b, respectively.

RESPONSE-We thank the reviewer for raising this question and apologize for the confusion. The GFP activation and C-to-T conversion rates were obtained based

on different cell populations. GFP activation was calculated based on sorted cells while C-to-T conversion was calculated based on the unenriched whole population of transfected cells. To avoid the confusion, we have adjusted *Fig. 1b* to normalized data and maintain the original gating information in *Supplementary Fig. 1d*. In addition, we have revised the Methods section to clearly indicate the experimental procedures.

In Fig. 1 legend, a sentence, “c Flow chart showing experimental procedures.”, should be removed. It would be better to explain Fig. 1b and 1c separately, and more clearly.

RESPONSE-We thank the reviewer for pointing this out. We have corrected the figure legend and revised the text in the Results and Methods sections to better describe these experiments.

In page 5 on line 9, what does “lowest inhibition rates” mean?

RESPONSE-We have revised the text to better describe the experimental results.

In Fig. 2a, the position 20 of *EPPK1* should be 12.

RESPONSE-We thank the reviewer to point this out and have corrected the figure.

Which date is Figure 2d based on? As the canonical targeting window of A3A CBE is 2 to 13 (described in Introduction of this manuscript), it would be better to show the inhibition data at position of 2, 4, 9, 10, and 13, or explain if the authors have any additional information.

RESPONSE-We apologize for the confusion. The data in *Fig. 2d* are derived from *Fig. 2c* and shown as schematic presentation. With regards to the targeting window, the canonical window of CBEs is deemed as positions 4 to 8 and A3A CBE has broadened window from 2 to 13. We have revised relevant text and figure legend to avoid confusion.

In Fig. 4b and Supplementary Fig. 1, the Sanger sequence results are too small to see the differences. Supplementary Fig. 1d showed 34.7% double positive cells for A3A CBE, but the G-to-A conversion is not seen with such ratio in the corresponding Sanger sequencing result. Generally, quantifying mixed populations by Sanger sequencing is difficult and the authors need to describe in detail how they ensure the quantitiveness.

RESPONSE-We have revised *Fig.4b* to improve the resolution of Sanger sequencing. Quantification of the Sanger sequencing results was performed as described (Kluesner *et al.*, 2018, *CRISPR J*). However, we agree with the reviewer that next-generation sequencing (NGS) is a better approach to quantify nucleotide mutations. Thus, we performed NGS analysis of the original data and have replaced the Sanger sequencing results in *Fig. 1c* with NGS results. Correspondingly, the Sanger sequencing results in *Supplementary Fig. 1e* are removed. It must be noted that the NGS results are generally consistent with the Sanger sequencing results and do not affect the conclusion of this study.

With regards to the discrepancy between GFP activation and nucleotide mutation, we have provided explanation in the above to indicate that these two quantification methods were calculated based on different cell populations.

Fig. 4f is not properly labeled.

RESPONSE-We thank the reviewer to point this out and have corrected the labeling in Fig. 4f.

Sup Fig 5 shows almost no effect on some target. More targets need to be analyzed to assess generality.

RESPONSE-We have included additional genomic sites with high editing efficiency in Supplementary Fig. 5 to address the reviewer's concern. The results are consistent with previous findings.

In Sup Fig. 6b, y-axis should be A-to-G conversion for ABE

In Sup Fig. 6d, the scale of y-axis is wrong.

RESPONSE-We thank the reviewer to point these out and have corrected the labeling in the Supplementary Fig. 6b and d.

Reviewer #2 (Remarks to the Author):

Base editing is a powerful method for inducing a single nucleotide conversion without DNA double-strand break. In this manuscript, Jia et al. presented a method of using a bacteriophage-derived peptide, termed G8Ppd, for fine tuning of the editing window of base editors (BEs) in human cells. They were able to achieve a precise base editing to correct a Marfan syndrome mutation by temporal expression of the phage peptide. I have the following points to improve this manuscript.

1. In Fig. 3d-e, the authors provided the inhibitory activity of G8Ppd for ABEs at five genomic sites. How about the inhibitory activity for CBEs at the same sites?

RESPONSE-We thank the reviewer for raising this interesting question. Considering both the editing efficiency and nucleotide composition, we analyzed the effects of G8P_{PD} on the editing activity of A3A CBE at the *VISTA* site. It was found that G8P_{PD} could consistently inhibit the editing efficiency at both on-target and out-of-window positions. The new results have been provided as Fig. 3g.

2. According to the experimental scheme in Fig. 1e, the phage-derived peptide is transfected 24 hours before cells are transfected with BEs. A serial transfection is cumbersome and may not be feasible with certain cell types such as haematopoietic stem cells. I wonder whether the authors have tried co-transfection. Alternatively, the phage-derived peptide may be fused to BEs using a linker or a self-cleaving 2A peptide.

RESPONSE-We thank the reviewer for raising this important question. We performed additional experiments to analyze the effects of co-transfected G8P_{PD} on A3A CBE. The new results are provided as *Supplementary Fig. 10*. It was found that pre-transfection, but not co-transfection, of G8P_{PD} could improve the targeting specificity of A3A CBE. These results suggested that timed delivery of G8P_{PD} was important for the function of G8P_{PD}.

In fact, Reviewer 1 has similar concerns on the feasibility of using G8P_{PD} serial transfection for practical applications. As suggested by Review 1, we investigated the effects of G8P_{PD} on embryonic base editing. We co-injected G8P_{PD} mRNA and A3A CBE plasmids into mouse embryos and found that the use of G8P_{PD} could facilitate the generation of blastocysts with perfectly edited gene alleles (*Fig. 5* and *Supplementary Fig. 11*). We hope that the results from embryonic base editing with single transfection could address, at least in principle, the reviewer's concerns.

With regards to the genetic fusion of G8P_{PD}, we designed A3A CBE-G8P_{PD} fusion which are spaced by a P2A self-cleaving peptide. Our preliminary results suggested that G8P_{PD} fusion did not improve the targeting specificity of A3A CBE as G8P_{PD} plasmid did. Due to the specific mechanism of action of G8P_{PD} (Cui *et al.*, 2020, *Genome Biol*), which functions as allosteric inhibitors of SpCas9, the structural organization of the fusion protein may need to be further optimized to achieve beneficial effects. Our results thus represent preliminary attempts and do not reflect exhaustive efforts for the fusion protein. We have included these results in the File for Review Only to prevent misleading the readers and would like to address the fusion protein in future studies with more thorough and comprehensive analysis.

Minor points:

1. In the Fig. 1 legend, the authors should delete “c flow chart showing experimental procedures”, which was repeated in Fig. 1d.

RESPONSE-We thank the reviewer to point this out and have corrected the figure legend.

Reviewers' Comments:

Reviewer #1:

Remarks to the Author:

In the revised manuscript, the authors performed and added several experiments in response to the reviewers' comments. Although this manuscript has been significantly improved in several respects through the efforts of the authors, this reviewer still has concerns about its lack of generality and applicability, as discussed below.

As a proof of concept, the embryo-injection study is not convincing that their approach is useful over conventional methods. In general, RNP or RNA injection of all genome editing components is preferred because it reduces mosaicism (and possibly indel and bystander mutations by base editing) while retaining high editing efficiency. To see if their approach would be more preferred, it is to be compared with simple CBE RNA injection. More statistically reliable scale and reproductivity are also needed.

In Supplementary Fig. 7, the amounts of plasmid DNA are rather small (5 to 10-fold reduction?) compared to other experiments as described in Methods, and thus this reviewer wonders how these amounts were chosen and to be compared with other experiments. This reviewer also wanted to see titrated RNA transfection of CBE only and its effect on the activity window for comparison.

To gain its generality and applicability, the authors are also expected to suggest their standard method that includes the amount of DNA/RNA if it matters, and the expected outcome which is experimentally supported.

Please make the deposited data to SRA accessible for reviewers.

Reviewer #2:

None

Reviewer #1 (Remarks to the Author):

In the revised manuscript, the authors performed and added several experiments in response to the reviewers' comments. Although this manuscript has been significantly improved in several respects through the efforts of the authors, this reviewer still has concerns about its lack of generality and applicability, as discussed below.

As a proof of concept, the embryo-injection study is not convincing that their approach is useful over conventional methods. In general, RNP or RNA injection of all genome editing components is preferred because it reduces mosaicism (and possibly indel and bystander mutations by base editing) while retaining high editing efficiency. To see if their approach would be more preferred, it is to be compared with simple CBE RNA injection. More statistically reliable scale and reproductivity are also needed.

Response: We agree with the reviewer that RNP or RNA injection is more widely used. Thus to improve the reliability of our findings, we performed additional microinjection experiments using both A3A and BE3 CBEs along with sgRNA in the format of coding RNA (Fig. 5d and Supplementary Fig. 12-13). The results revealed consistent effects of G8P on BEs with previous experiments. In addition, to rule out the possibility that the lack of perfectly edited blastocysts in the absence of G8P_{PD} was due to the limited sample size, we analyzed 30 gene-modified blastocysts in the A3A mRNA minus G8P group and found that none of these blastocysts contained perfectly edited gene alleles.

In Supplementary Fig. 7, the amounts of plasmid DNA are rather small (5 to 10-fold reduction?) compared to other experiments as described in Methods, and thus this reviewer wonders how these amounts were chosen and to be compared with other experiments. This reviewer also wanted to see titrated RNA transfection of CBE only and its effect on the activity window for comparison.

To gain its generality and applicability, the authors are also expected to suggest their standard method that includes the amount of DNA/RNA if it matters, and the expected

outcome which is experimentally supported.

Response: We apologize for the confusion and thank the reviewer for pointing out these typos. The amount of BE and sgRNA were indicated in the cell culture and transfection section in Methods. During the whole studies, the conditions of transfection including the amount of DNA or RNA were kept consistent unless noted otherwise. We have corrected the typos in Supplementary Fig. 7 and revised the Methods to clearly indicate this.

Please make the deposited data to SRA accessible for reviewers.

Response: We have updated the SRA code for embryo sequencing in the Data Availability section.